

# Λ(1520) resonance production with respect to transverse spherocity using EPOS3+UrQMD

**Nasir Mehdi Malik⋆ and Sanjeev Singh Sambyal**

University of Jammu

⋆ nasir.mehdi.malik@cern.ch

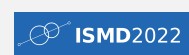

## Abstract

**Resonances are sensitive to the properties of the medium created in heavy ion collision. They also provide insight into the properties of the hadronic phase. Studying the dependence of the yield of resonances on transverse spherocity and multiplicity allows us to understand the resonance production mechanism with event topology and system size, respectively. The results reported pertain to Λ(1520), using predictions from EPOS3+UrQMD event generator.**

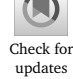
## 1 Introduction

At collider experiments, a major challenge is to understand production/interaction of resonances in a hot dense medium because their lifetime is comparable to that of hadronic phases [1]. Resonance particle Λ(1520) having a life-time between $K^{*0\pm}(980)$ and $\phi(1020)$ make it a good candidate to understand the hadronic phase [2, 3]. Here we are analysing production of Λ(1520) with respect to transverse spherocity $S_0$ using the EPOS3 event generator [4–6] with UrQMD as an afterburner to simulate the hadronic rescatterings after hadronisation.

Transverse spherocity is an event shape variable which help to distinguish between isotropic and jetty events. Its value ranges from 0 to 1 and it is given by

$$S_0 = \frac{\pi^2}{4} \min_{\vec{n}=(n_x,n_y,0)} \left( \frac{\sum_i \left| \vec{P}_{T_i} \times \hat{n} \right|}{N_{\text{particles}>5}} \right)^2,$$

where

$$S_0 = \begin{cases} 0, & \text{``pencil-like'' limit (hard events),} \\ 1, & \text{``isotropic'' limit (soft events),} \end{cases}$$

Table 1

(a) Percentile of multiplicity distribution.

| No. of charged particle | Percentile (lower) | Percentile (upper) |
|---|---|---|
| 28 - 110 | 90-100% | 0-10% |
| 18-28 | 70-90% | 10-30% |
| 11 - 18 | 40-70% | 30-60% |
| 0-110 | 0-100% | 0-100% |

(b) Quantile of spherocity distribution.

| Multiplicity(%) | Jetty 0-10% (lower) | Isotropic 0-10% (upper) |
|---|---|---|
| 0-100 | 0.00-0.381 | 0.834-1.00 |
| 0-10 | 0.00-0.644 | 0.896-1.00 |
| 30-60 | 0.00-0.442 | 0.824-1.00 |

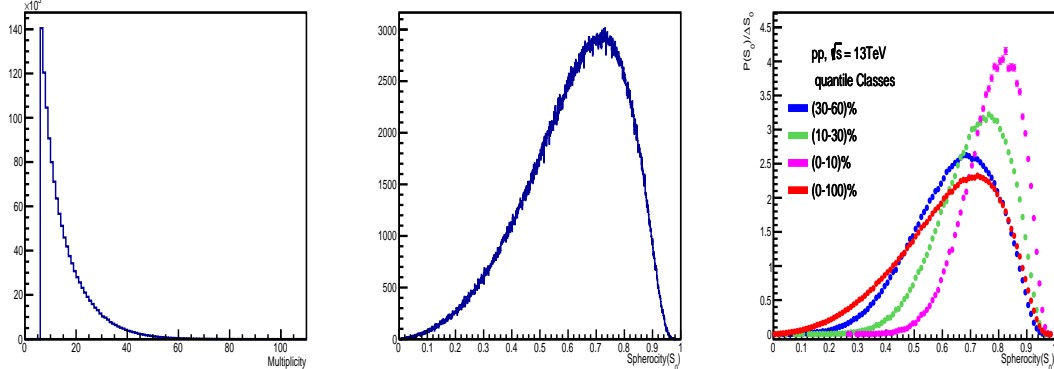

Figure 1: Shows distribution of: (a) Charge particle multiplicities. (b) Transverse spherocity. (c) Transverse spherocity vs multiplicity.

and $\hat{n}$ is a two dimensional unit vector in the transverse plane,chosen such that $S_0$ is minimised. By restricting it to the transverse plane, transverse spherocity becomes infrared and collinear safe. Also the jetty events are usually hard events while the isotropic ones are the result of soft processes [7–10].

## 2 Distribution plots

For the present study, events having more than 5 charge paricles are considered. Moreover, to study in various multiplicity intervals and transverse spheorcity classes, we separated events into the 0-10% quantile(high multiplicity) and 30-60% quantile(low multiplicity). To disentangle the spherocity classes i.e jetty and isotropic events, we have applied event-selection cuts on the spherocity distribution. The values, cuts and definition of various multiplicity percentile and spherocity classes can be found in Table 1a and 1b. We observed that the spherocity distributions are shifted toward 1 from the low multiplicity quantile to the high multiplicty one. Figure 1 displays charged particle multiplicity, transverse spherocity and spherocity as a function of multiplicity.

## 3 pT spectra of Λ(1520)

Figure 2(2a,2b) shows the pT-spectra of Λ(1520) for 0-10% and 30-60% multiplicity classes with top and bottom 10% spherocity values responsible for selecting jetty and isotropic events respectively. The bottom panels show the ratio with respect to the spherocity integrated class.

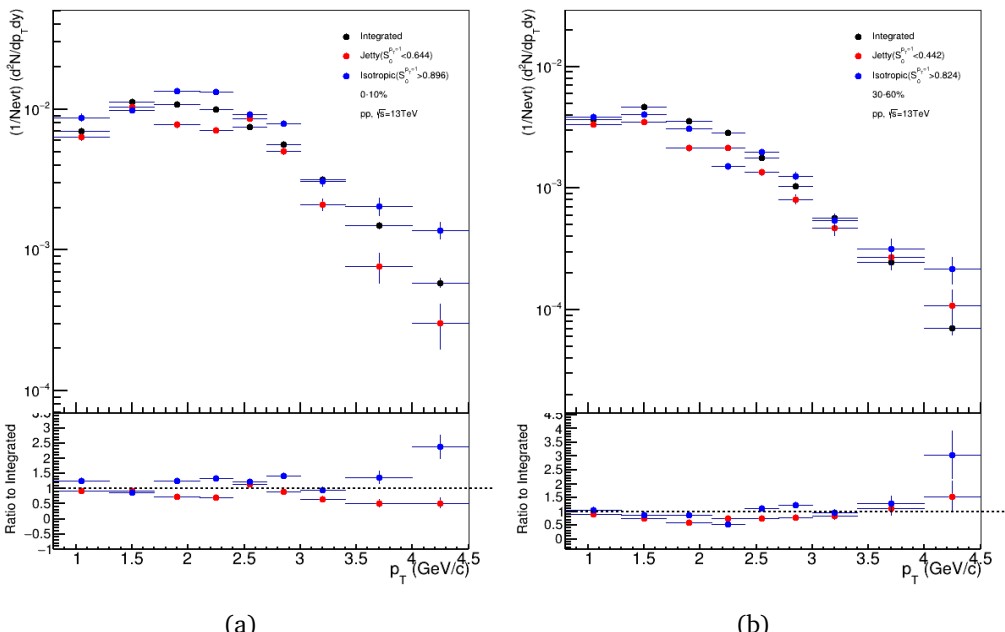

Figure 2: (a) pT spectra of $\Lambda(1520)$ evaluated at 0-10% multiplicity for Isotropic(blue), Jetty (red) and Integrated(black). (b) pT spectra of $\Lambda(1520)$ evaluated at 30-60% multiplicity for Isotropic(blue), Jetty (red) and Integrated(black).

## 4 Conclusion

A study of the transverse spherocity of $\Lambda(1520)$ in p-p collisions at $\sqrt{s} = 13$ TeV using EPOS3 generator with UrQMD as afterburner was reported. The observables are measured using primary charge particles and reported as a function of charged particle multiplicity for events with different scales defined by the transverse spherocity. As can be seen in Figure(1c),jetty events dominate the low-multiplicity region. However, we see that production of $\Lambda(1520)$ is dominated in both multiplicity regions by isotropic event shapes, since EPOS is a parton model, with many parton-parton binary interaction,so highest multiple interaction(MPI) mean dominace in isotropic event(soft QCD). An experimental investigation in this direction would be very beneficial to comprehend the event shape dependence of system dynamics.

## Acknowledgements

We thank Vikash Sumberia, Randhir Singh and Dr. Ranbir Singh for inspiring us to do this work and providing also valuable advice. Thanks are also due to Johannes Jahan Ph.D. student in Subatech, Nantes -France for helping us to understand technical term in EPOS3.

**Funding information** This work was supported by Council of Scientific and Industrial Research (CSIR), India. The authors acknowledge the HEP Lab. Department of Physics, University of Jammu for providing infrastructure for computing etc.

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
