# Peer review of "Λ(1520) resonance production with respect to transverse spherocity using EPOS3+UrQMD"

_SciPost Physics Proceedings, doi:SciPost Phys. Proc. 15, 027 (2024)_

## Round 1 · Referee Report · Anonymous (Referee 1) · 2023-2-28

Report

The presentation could do with some improvement, as it features multiple mis-spellings and grammatical issues: fortunately these should be easy to fix, and suggestions are made below.

The overall message of the paper lacks both a comparison to any real data, or a discussion of *why* the EPOS+UrQMD generator model would favour Lambda production in isotropic event types. This would be a good short discussion to add along with the results presentation or in the conclusions.

Requested changes

  1. Abstract: "The results reported pertains... The data from EPOS3 is used for the present analysis." -> "The results reported pertain to Lambda(1520), using predictions from the EPOS3+UrQMD event generator." The results should not be described as EPOS3 alone, as UrQMD is used and may influence the results: this would also motivate a change of title to reflect use of UrQMD unless it is really not important and all the same features are present with standalone EPOS3.

  2. p1, "In collider experiment , people are trying to understand..." -> rephrase and correct spacing, e.g. "At collider experiments, a major challenge is to understand..."

  3. "because they have life-time comparable to" -> "because their lifetime is comparable to"

  4. "using event-generator EPOS3 ... as afterburner" -> "using the EPOS3 event generator ... as an 'afterburner'". This second part needs expansion to state what UrQMD actually does, and the meaning of "afterburner": it would be better to avoid this jargon.

  5. "distinguish isotropic and jetty event" -> "distinguish between isotropic and jetty events"

  6. "ranges form" -> "ranges from"

  7. First equation: use \mathrm or text for upright text and operators, such as "min" and "particles" here. The following "Where," should be "\noindent where" with comma at the ends of both equations to make coherent sentences.

  8. "chosen in a way such that" -> "chosen such that"

  9. Better to put all the consecutive \cite commands together so they will be grouped by BibTeX, e.g. "\cite{A}\cite{B}..." -> "\cite{A,B,...}"

  10. Sec 2, "events having charge paricles greater than 5" -> "events having more than 5 charged particles"

  11. "Moreover, to study in various multiplicity interval and transverse spheorcity classes, First we took the quantile of multiplicity distribution in two regions 0-10%(high multiplicity) and 30-60%(low multiplicity)." -> "Moreover, to study in various multiplicity intervals and transverse-spherocity classes, we separated events into the 0-10% multiplicity quantile (high multiplicity) and 30-60% quantile (low multiplicity)."

  12. "disentagle" -> "disentangle"

  13. "we have applied spherocity cuts on spherocity distribution" -> "we have applied event-selection cuts on the spherocity distribution"

  14. "from low multiplicity percentile to high multiplicty" -> " from the low multiplicity quantile to the high multiplicity one".

  15. Figure 1 caption: " spherociyty" -> "spherocity"

  16. Figure 2 caption: " Isotrophic" -> "Isotropic" (twice)

  17. Sec 4, "13 TeV" needs the TeV upright, e.g. via \mathrm

  18. URQMD: capitalise consistently with earlier UrQMD, again avoid or explain "afterburner". "Post-processing" might be clearer.

  19. "measured using primary charge particles" -> "measured using charged primary particles"

  20. "spehrocity" -> "spherocity"

  21. "As it we can be seen in Figure(1c), that jetty events is dominating in low multiplicities" -> "As can be seen in Figure(1c), jetty events dominate the low-multiplicity region."

  22. "When comparing 0-10% and 30-60% multiplicity class ,but we see that production of Λ(1520) in isotropic event shape is dominating in both cases" -> "However, we see that production of Λ(1520) is dominated in both multiplicity regions by isotropic event shapes." Scientifically, I don't think this correlation is clearly shown, and there is a lot of statistical uncertainty from the modelling that make "dominance" not so clear - isotropic production does appear to be larger, though.

---

## Editorial Decision

published